# The Relation between Plasma Nesfatin-1 Levels and Aggressive Behavior in Pit Bull Dogs

**DOI:** 10.3390/ani14040632

**Published:** 2024-02-16

**Authors:** Gokcen Guvenc-Bayram, Zeynep Semen, Pelin Fatos Polat-Dincer, Zeynep Tugce Sertkaya, Yasemin Ustundag, Can Ates, Bugra Aktas, Murat Yalcin

**Affiliations:** 1Department of Physiology, Faculty of Veterinary Medicine, Dokuz Eylul University, Izmir 35890, Turkey; 2Department of Biochemistry, Faculty of Veterinary Medicine, Dokuz Eylul University, Izmir 35890, Turkey; zeynep.semen@deu.edu.tr; 3Department of Internal Medicine, Faculty of Veterinary Medicine, Dokuz Eylul University, Izmir 35890, Turkey; pelinfatos.polat@deu.edu.tr; 4Department of Physiology, Faculty of Medicine, Ankara Medipol University, Ankara 06050, Turkey; zeynep.sertkaya@ankaramedipol.edu.tr; 5Department of Anatomy, Faculty of Veterinary Medicine, Dokuz Eylul University, Izmir 35890, Turkey; yasemin.ustundag@deu.edu.tr; 6Department of Biostatistics, Faculty of Medicine, Aksaray University, Aksaray 68100, Turkey; can.ates@gmail.com; 7Manisa Metropolitan Municipality Temporary Animal Shelter, Manisa 45125, Turkey; vethekaktas@gmail.com; 8Department of Physiology, Faculty of Veterinary Medicine, Bursa Uludag University, Bursa 16059, Turkey; muraty@uludag.edu.tr

**Keywords:** Pit Bull dog breeds, aggression, nesfatin-1, serotonin, oxytocin, dopamine

## Abstract

**Simple Summary:**

Aggression stands out as one of the most prevalent behavior problems in dogs, with a particular predisposition noted in Pit Bull breeds. To advance our understanding and develop new treatment strategies for aggression, it is crucial to explore the neurobiological foundations of this behavior. In our study, Pit Bull dogs were categorized based on aggression levels through the administration of aggression tests. Plasma nesfatin-1, serotonin, dopamine, and oxytocin levels were determined in blood samples collected after a 24 h fasting period and 2 h after refeeding. The results, a pioneering contribution to the literature, indicate a novel association between decreased plasma levels of nesfatin-1, serotonin, and oxytocin and increased dopamine levels with aggression in Pit Bull dogs. During fasting conditions, lower plasma levels of nesfatin-1, serotonin, and dopamine were observed, while plasma oxytocin levels were higher. Our findings underscore the significant roles of nesfatin-1, serotonin, oxytocin, and dopamine in the aggression of Pit Bull dogs, revealing potential interactions among these neuropeptides at the central nervous system level.

**Abstract:**

Aggression is a prevalent and concerning behavioral issue in dogs. Pit Bull dogs, known for their high levels of aggression, are recognized as a focus of concern in society. In our study, we aimed to investigate the behavioral characteristics of Pit Bull dogs and explore the potential roles of peptides involved in the neurobiology of aggression. Initially, female, and male dogs underwent aggression tests, and their aggression levels were categorized. Plasma nesfatin-1, serotonin, oxytocin, and dopamine levels were quantified using ELISA, with blood samples collected after a 24 h fasting period and 2 h post-refeeding. Our findings indicate that aggression in Pit Bull dogs correlates with decreased plasma nesfatin-1, serotonin, and oxytocin levels, while dopamine levels increase. The study’s findings indicate that fasted dogs exhibited lower plasma levels of nesfatin-1, serotonin, and dopamine, while plasma oxytocin levels were higher. Furthermore, while the research findings do not suggest a significant relationship between the severity of aggression and the gender of the dog, male Pit Bull breeds appear to have higher plasma nesfatin-1 and serotonin levels compared to their female counterparts. The study’s findings demonstrate that nesfatin-1, serotonin, oxytocin, and dopamine play pivotal roles in Pit Bull dogs’ aggression, indicating potential interactions among these neuropeptides at the central nervous system level.

## 1. Introduction

Animal behavior, a multifaceted phenomenon, encompasses an array of survival behaviors intricately shaped by interactions with the natural environment [1]. In particular, domesticated dogs (*Canis lupus familiaris*) have exhibited notable behavioral changes compared to their wolf ancestors (*Canis lupus*), primarily due to their close association with humans throughout the process of domestication. This evolutionary journey has led to significant adaptability in dogs, aligning their behavior more closely with human society [2]. However, this rapid and profound shift in living environments has not been fully paralleled by corresponding biological adaptations. Consequently, many dogs experience internal conflicts arising from the mismatch between their inherent biological predispositions and the demands of their new environments. This dissonance often manifests as various stress-related behaviors, including aggression, anxiety, and depression-like symptoms [3,4]. These behavioral patterns, when deemed undesirable or problematic by humans, are often labeled as behavioral disorders [5]. These behavioral disorders in dogs can manifest due to various factors, including genetic diseases, developmental challenges, breed-specific behaviors, social environment issues, adaptation strategies, and environmental adaptation disorders [1].

Aggression in dogs is a prevalent and hazardous behavioral problem. It poses significant risks, both to the safety of humans and to the welfare of the dogs themselves, often leading to unfortunate outcomes such as harm to humans or euthanasia of the aggressive dogs [6,7]. The etiology of canine aggression is multifaceted, encompassing medical origins such as physical ailments, endocrine system disorders, infectious diseases, central nervous system disorders, and hereditary conditions [8,9,10]. Furthermore, individual differences, including breed-specific predispositions and personal temperament, also play a crucial role in the manifestation of aggression in dogs [11].

The classification of aggression in dogs is diverse, encompassing various types based on their underlying causes. These include aggression motivated by fear and self-defense, territorial aggression, aggression related to herd dynamics, and aggression influenced by racial and hereditary factors. Other types include aggression observed during the socialization period, aggression stemming from diseases, dominant aggression, and aggression of unknown origins [12,13,14,15]. Behavioral tests encompass various criteria for identifying dogs predisposed to dominant aggression. These criteria involve a range of interactions and responses, including reactions to eye contact, routine care procedures, food-related interactions, physical contact, therapeutic procedures, disciplinary actions, leash handling, and responses to strangers. Dogs that display an aggressive attitude towards at least five of these criteria are classified as predisposed to dominant aggression [9,10,16,17,18]. Notably, research has shown a higher prevalence of dominant aggression in young, male, purebred dogs, with certain breeds like bull terriers showing a higher incidence of such behaviors [19,20]. Additionally, studies have consistently observed that Pit Bull breeds exhibit higher levels of aggression compared to other breeds [11,21].

Pit Bull breeds, characterized by their muscular build and robust nature, hold a dual image in society. On one hand, they are valued for attributes such as agility, bravery, protectiveness, and loyalty, making them desirable pets for some. On the other hand, they are often viewed with apprehension due to their historical involvement in aggressive activities and negative media portrayal [22,23,24]. The Pit Bull breeds, including the American Pit Bull Terrier, the American Bulldog, the American Staffordshire Terrier, and the Staffordshire Bull Terrier, originated in England through the crossing of Terrier and Bulldog breeds. Historically, Pit Bulls have been utilized in various roles; their warrior-like and aggressive traits initially made them popular in blood sports such as dog fighting, cock fighting, gladiator fights, hunting, and bullfighting, and they symbolized American resilience during the World Wars. Their association with such aggressive activities, combined with their portrayal in the media, has significantly influenced public perception of these breeds. Despite these challenges, Pit Bulls have also been valued as “nanny dogs” due to their loyalty, obedience, affection, and protectiveness [23,24,25,26]. Nevertheless, the negative stereotypes surrounding Pit Bulls, stemming from their historical association with illegal dog fights and sensationalized media coverage, have exacerbated the challenges of addressing aggression issues in these breeds. While research suggests that appropriate training and socialization could mitigate aggression in dogs, these negative stereotypes continue to influence public opinion and policy [25,26]. Despite the critical importance of this issue, a notable gap in the scientific literature exists regarding the investigation of the underlying causes of aggression in dogs, particularly with a specific scarcity of research focusing on understanding the origins of this behavior within the Pit Bull breed.

Serotonin and dopamine are neurotransmitters with significant roles in various behavioral and neuropsychological processes, including emotional regulation and cognitive functions in mammals [27,28,29,30,31]. Low serotonin levels have been correlated with aggressive and impulsive behaviors in both humans and animals [32,33,34,35,36], while elevated levels of serotonin are associated with anxiety disorders [37,38,39]. Genetic studies have established a connection between the dopaminergic system and aggression, highlighting a link between aggression and a dopamine D4 receptor polymorphism [40,41]. Consequently, this suggests that the dopaminergic system may serve as a potential biomarker for aggression tendencies in mammals [29].

Oxytocin, a neuropeptide known for its involvement in social bonding [42], has been associated with psychiatric conditions [43]. While studies suggest that oxytocin levels in dogs may be affected by their proximity to their owners [44,45], the exact mechanisms behind this phenomenon remain elusive.

Nesfatin-1, another neuropeptide, participates in various physiological processes and interacts with the serotonin, dopamine, and oxytocin systems [46,47]. While its effect on dog aggression remains unexplored, correlations have been observed between nesfatin-1 and stress responses, anxiety, and depression-like behaviors in humans and rodents [48,49,50,51,52]. Given nesfatin-1’s interactions with other neurotransmitter systems and its impact on hunger and satiety [47,53], it holds relevance for comprehending canine aggression.

Considering all of this information, the current study, based on the roles of serotonin, oxytocin, and dopamine in aggressive behavior, as well as the interaction of nesfatin-1 with these neuropeptides, hypothesizes a significant difference in plasma nesfatin-1 levels between aggressive and non-aggressive Pit Bull individuals. To test this hypothesis, our study aims to compare nesfatin-1 hormone levels between aggressive and non-aggressive Pit Bull dogs. Additionally, we intend to examine alterations in plasma serotonin, dopamine, and oxytocin levels in aggressive and non-aggressive Pit Bulls and explore the relationships between changes in plasma levels of these neuropeptides and plasma nesfatin-1 levels depending on aggression.

## 2. Materials and Methods

### 2.1. Animals and Grouping of Animals by Aggression

A total of 66 Pit Bull dogs housed at the Manisa Metropolitan Municipality Temporary Animal Shelter, comprising 35 females and 31 males aged between 1 and 5 years, were included in this study. Dogs presumed to be Pit Bulls were included in the study based on their phenotypic characteristics, such as large skulls; pronounced muscular build, especially in the hindquarters; broad heads and jaws; tight skin; etc. [24]. The dogs were individually housed in standard cages with access to regular feeding and water. A comprehensive clinical examination of each dog was conducted to ensure the inclusion of only healthy individuals in the study. This examination included a general physical assessment, evaluation of vital signs, and screening for any signs of infectious diseases or chronic health issues. Dogs with any known chronic illnesses or health conditions that could potentially influence behavior were excluded from the study.

To assess aggression levels, individual aggression tests were administered by a veterinarian who routinely cared for the shelter dogs. These tests were conducted under standard conditions to ensure the welfare of the dogs [17,54]. Before commencing the aggression tests, a detailed informational form was completed for each dog. This form included general information such as age, gender, date of arrival at the shelter, any known chronic illnesses, history of biting, involvement in dog fighting, incidents of known violence towards staff, and body mass index (BMI). The aggression test itself was adapted from the Canine Behavioral Assessment and Research Questionnaire (C-BARQ), but tailored to the specific conditions of the shelter to minimize stress and potential biases (Appendix A). The test involved a series of behavioral observations and interactions, covering various scenarios such as the dog’s behavior in terms of protecting its food; guarding its territory; and reacting to perceived threats, both direct and indirect. The dog’s behavior towards loved ones and strangers was also assessed. A scoring system, ranging from 1 (Never) to 4 (Always), was employed to quantify observed behaviors. This system was used to objectively measure responses such as remaining calm, growling, barking, displaying teeth, making distant biting gestures, and biting with intent. The cumulative scores from these tests were used to categorize the dogs into three distinct groups based on their aggression levels: non-aggressive (NA), aggressive (A), and dominant aggressive (DA). The non-aggressive group (NA) included dogs with scores ranging from 22 to 44, typically exhibiting quiet and calm behavior. The aggressive group (A) comprised dogs with scores between 44 and 66, characterized by behaviors such as growling, barking, and displaying teeth. The dominant aggressive group (DA) consisted of dogs scoring between 66 and 88, notably displaying more severe behaviors like distant biting gestures and biting with the intent to attack.

### 2.2. Experimental Protocols

All dogs underwent a 24 h fasting period with access to water to evaluate their hunger and satiety statuses. Blood samples were collected before and 2 h after refeeding using appropriately sized muzzles. During the blood sampling, a tourniquet was applied 2 cm above the left forearm’s elbow joint, and 2 mL of blood was drawn from the cephalic vein using an appropriate needle after cleaning the area. The collected blood samples were promptly placed on ice. After centrifugation at +4 °C and 1800 rpm for 20 min, plasma was divided into four equal volumes to measure plasma concentrations of nesfatin-1, serotonin, dopamine, and oxytocin, and then stored at −80 °C until further analysis.

### 2.3. Determination of Plasma Nesfatin-1, Serotonin, Dopamine, and Oxytocin Levels

The plasma concentrations of nesfatin-1, serotonin, dopamine, and oxytocin were determined using ELISA kits obtained from Bioassay Technology Laboratory (BT-Lab, Shanghai, China), with the following catalog numbers: nesfatin-1 (catalogue number: E0141Ca), serotonin (catalogue number: E0263Ca), dopamine (catalogue number: EA0020Ca), and oxytocin (catalogue number: EA0053Ca). The measurements were conducted in accordance with the kit manuals.

In brief, 50 μL of plasma samples were added to microplate wells pre-coated with antibodies specific to canine nesfatin-1, serotonin, dopamine, or oxytocin. After an incubation period and subsequent washing steps, biotin-conjugated antibodies for each analyte were added. Following another incubation and washing cycle, streptavidin-HRP was introduced into the wells to form an immune complex. After further incubation and washing to eliminate unbound enzymes, a chromogenic HRP enzyme substrate solution was applied to the wells, resulting in a color change to blue. To terminate the HRP enzyme reaction, a stop solution was added to the wells. The absorbance of the plates was measured at 450 nm using a plate reader (Allsheng.AMR-100). The kits provided detection limits for nesfatin-1, serotonin, dopamine, and oxytocin at 0.3–90 ng/mL, 10–640 ng/mL, 5–1000 ng/L, and 0.75–48 ng/L, respectively.

### 2.4. Data and Statistical Analyses

The study’s a priori sample size was determined based on the aim of detecting a significant difference of 25% in the means of Nesfatin-1, the primary outcome variable, between groups. To achieve this, a one-way analysis of variance (two-sided hypothesis test) with a minimum power of 80% and a type 1 error level of 0.05 required a total of 60 subjects. Considering a potential dropout rate of 10%, the study was completed with a total of 66 animals [55]. Data analysis was performed using IBM SPSS Statistics for Windows, version 25.0. To assess differences in variables such as plasma nesfatin-1, serotonin, dopamine, and oxytocin between groups (non-aggressive, aggressive, and dominantly aggressive), an analysis of variance (ANOVA) was employed, followed by post hoc multi-comparison tests (Tukey’s HSD) to identify specific groups with significant differences. Paired-t tests were used to compare the same dogs when fasted and after feeding, within their respective groups. Descriptive statistics are presented as the mean ± standard deviation for continuous variables. Bar graphs with standard errors were used to illustrate mean differences in related variables. A “*p*” value of less than 0.05 was considered statistically significant [56].

## 3. Results

In the categorization based on the aggression test results, the non-aggressive (NA) group included 17 dogs, accounting for 25.76% of the total cohort. This group comprised eight males (12.12% of the total, representing 47.06% of the NA group) and nine females (13.64% of the total, 52.94% of the NA group). The aggressive (A) group consisted of 29 dogs, making up 43.94% of the cohort, with 10 males (15.15% of the total, 34.48% of the A group) and 19 females (28.79% of the total, 65.52% of the A group). The dominant aggressive (DA) group included 20 dogs, which was 30.30% of the total population, with 13 males (19.70% of the total, 65% of the DA group) and 7 females (10.61% of the total, 35% of the DA group).

The plasma nesfatin-1 levels of Pit Bulls in the NA, A, and DA groups showed an increase following refeeding compared to their fasting levels. Notably, a trend of decreasing plasma nesfatin-1 levels was observed in correlation with the degree of aggression in both fasting and refeed states. Additionally, male dogs in all groups tended to exhibit higher plasma nesfatin-1 levels compared to their female counterparts (Figure 1).

Upon refeeding the animals after a 24 h fasting period, an increase in plasma serotonin levels was observed across all groups. Notably, aggressive and dominant aggressive dogs of both sexes showed lower plasma serotonin levels compared to non-aggressive dogs. Furthermore, male Pit Bulls in each group displayed higher plasma serotonin levels than females (Figure 2).

The expression of aggression seemed to coincide with an increase in plasma dopamine levels in both male and female Pit Bulls. Furthermore, refeeding the animals after fasting also tended to result in elevated plasma dopamine levels in all groups. No gender-related differences were observed in the plasma dopamine levels of the Pit Bulls within each group (Figure 3).

Finally, the plasma oxytocin levels of both male and female Pit Bulls in the A and DA groups were lower than those in the NA group. Refeeding after a 24 h fast did not significantly alter plasma oxytocin levels in the NA group, but tended to decrease them in the A and DA groups. There were no evident differences in plasma oxytocin levels between male and female dogs within these groups (Figure 4).

## 4. Discussion

Our study provides valuable insights into the complex relationship between plasma levels of nesfatin-1, serotonin, oxytocin, and dopamine and their associations with aggression in Pit Bull dogs. Key findings include a consistent decrease in plasma nesfatin-1, serotonin, and oxytocin levels in tandem with an escalation of aggression, coupled with an increase in plasma dopamine levels. Additionally, the fasting state appeared to influence these neurochemical levels, generally resulting in lower levels of nesfatin-1, serotonin, and dopamine in both aggressive and non-aggressive dogs. Conversely, fasting led to an increase in plasma oxytocin levels compared to when the dogs were fed. Although our statistical analysis did not establish a significant correlation between aggression severity and the dogs’ gender, an interesting pattern emerged: male Pit Bulls exhibited consistently higher plasma levels of nesfatin-1 and serotonin compared to females across all aggression categories.

One of the most striking observations from our research is the inverse relationship identified between the severity of aggression and plasma nesfatin-1 levels. This relationship suggests that, as aggression intensifies, there is a corresponding decrease in plasma nesfatin-1 levels in both male and female Pit Bulls. This inverse dynamic highlights a potential neurobiological link between aggression and nesfatin-1, a neuropeptide traditionally associated with appetite and stress responses, underscoring the multifaceted nature of canine aggression and its underlying physiological mechanisms. Nesfatin-1 is an anorexigenic neuropeptide expressed in the central nervous system and peripheral tissues [46,57,58]. It plays a crucial role in various physiological functions, including the regulation of food and water intake, energy metabolism, thermoregulation, the cardiovascular system, the modulation of emotional states such as anxiety and depression, and reproductive functions [59]. Furthermore, nesfatin-1 is believed to be involved in the regulation of stress responses and emotional behaviors in humans [60,61,62].

The evolutionary journey from wolves to domesticated dogs has resulted in significant behavioral adaptations. This process has allowed dogs to become closer to humans, but it has also led to potential conflicts between their inherent traits and the demands of a domestic environment [2]. Such conflicts can manifest as stress, aggression, anxiety, and behaviors resembling depression, often perceived as behavioral disorders in dogs [3,4]. Our study contributes to this understanding by examining these behaviors in Pit Bull dogs, a breed often associated with aggression.

Our application of the C-BARQ provided a structured approach to evaluating aggression [17]. In our cohort of 66 Pit Bull dogs, we observed a range of behaviors, with 17 dogs not exhibiting aggressive behavior, 29 displaying aggression, and 20 showing dominant aggressive characteristics. These behaviors, including food protection, territorial defense, and reaction to unfamiliar situations, highlight the varied nature of aggression in dogs depending on the stimulus intensity.

Achieving results similar to those of our study, Duffy et al. (2008) [21] categorized canine aggression based on targets, including owners, unfamiliar people, and other dogs. They noted that breeds perceived as friendly showed slight aggression towards strangers and other dogs, while breeds like Pit Bulls exhibited more pronounced aggression. Conversely, MacNeil-Allcock et al. (2011) reported that Pit Bulls adopted from shelters did not show higher aggression levels than other breeds, suggesting that environmental factors, such as upbringing and past experiences, significantly influence aggression [18].

Another study focused on investigating aggression levels based on scar size in pit bulls, a breed often associated with aggression and then euthanasia. This study’s findings suggest that Pit Bulls with larger scars, potentially indicative of past aggressive encounters, tend to display higher levels of aggression, particularly males. However, this approach, focusing solely on physical markers like scars, offers a limited view of a dog’s behavioral tendencies. Scars might reflect past experiences, but do not necessarily portray current temperament or behavioral predispositions. Behavioral assessments, such as those conducted using the C-BARQ, provide a more holistic and accurate measure of aggression by evaluating responses to diverse stimuli and situations [63]. It is crucial to consider both the observable physical indicators and the broader neurobiological and environmental factors when assessing canine aggression. This comprehensive approach ensures a nuanced understanding, moving beyond breed-specific stereotypes and acknowledging the complex interplay of genetic, neurochemical, and experiential factors in shaping a dog’s behavior [5].

Aggression in dogs can be classified into various types, each attributable to different causes [12,13,14,15]. Dominant aggression is the most common form and occurs independently of breed, gender, and age characteristics [20,32,64,65]. Studies have shown that dominant aggression is frequently observed in young male dogs and purebreds, notably in bull terriers [19,20]. To gain a deeper understanding of breed-specific aggression patterns, a comprehensive study using the C-BARQ was conducted, evaluating the behaviors of over 30 dog breeds in response to a variety of stimuli and situations [21,66]. This study revealed significant variations in aggression levels among different breeds. For example, Dachshunds, English Springer Spaniels, Golden Retrievers, Labrador Retrievers, Poodles, Rottweilers, Shetland Sheepdogs, and Siberian Huskies exhibited similar aggression levels towards strangers, other dogs, and their owners. Breeds like Chihuahuas and Dachshunds scored above average in aggression towards humans and dogs, while Akitas and Pit Bull Terriers showed high aggression levels towards specific targets, particularly other dogs. In contrast, breeds such as Golden Retrievers, Labrador Retrievers, Bernese Mountain Dogs, Brittany Spaniels, Greyhounds, and Whippets displayed relatively low aggression towards both humans and dogs [21].

The study also found that certain breeds, including Dachshunds, Chihuahuas, and Jack Russell Terriers, demonstrated a propensity for aggressive behavior towards specific groups, such as strangers and owners. Australian Cattle Dogs exhibited higher aggression towards foreigners, while American Cocker Spaniels and Beagles showed aggression towards their owners. Notably, more than 20% of Akitas, Jack Russell Terriers, and Pit Bull Terriers exhibited highly aggressive behavior towards unfamiliar dogs [67].

Studies have indicated that nesfatin-1 levels fluctuate in response to stress-related situations and psychiatric disorders, with higher plasma nesfatin-1 levels found in individuals with Major Depressive Disorder, which correlates positively with the severity of depression [62,68]. Additionally, nesfatin-1 has been considered a potential biomarker for depression and anxiety disorders [69,70]. Notably, a positive relationship between plasma nesfatin-1 levels and the severity of anxiety has been reported [71]. Patients diagnosed with alcohol abuse disorder exhibited notably lower concentrations of plasma nesfatin-1 when compared to healthy control subjects [61]. Similarly, during the manic periods of bipolar disorder, individuals displayed reduced plasma nesfatin-1 levels [72]. It is noteworthy that both bipolar disorder and alcohol abuse disorder are frequently associated with aggressive behaviors, including self-harm, impulsivity, and engagement in criminal activities [73,74]. Furthermore, individuals diagnosed with Antisocial Personality Disorder (ASPD), a condition characterized by heightened aggression, have been found to have diminished plasma nesfatin-1 levels in comparison to healthy individuals [75]. This discovery highlights a consistent trend linking lower nesfatin-1 levels with heightened aggression in various clinical contexts. The reduced plasma nesfatin-1 levels observed in dogs in our current study, correlating with the severity of aggression, bear a striking resemblance to the diminished plasma nesfatin-1 levels observed in human patients diagnosed with bipolar disorder, alcohol abuse disorder, and ASPD, all of which are frequently characterized by heightened aggressive tendencies.

Previous research has indicated that nesfatin-1 secretion naturally diminishes during periods of fasting and starvation [76]. Our study aligns with this trend, revealing lower plasma nesfatin-1 levels in both non-aggressive and aggressive dogs during fasting compared to well-fed dogs, with the severity of aggression influencing these levels. These findings suggest a potential link between nesfatin-1 secretion and the manifestation of aggressive behavior, particularly in the context of fasting or hunger [77].

Although there was no statistically significant difference between the dogs’ aggression scores and gender in the current study, the plasma nesfatin-1 level of female dogs in all study groups were found to be lower than male dogs. These findings align with previous observations in healthy sheep and lambs, where males exhibited higher plasma nesfatin-1 levels than females [78]. Additionally, gender-related differences in plasma nesfatin-1 levels have been reported in individuals with various psychiatric diseases. Notably, as anxiety levels increase in women, nesfatin-1 levels tend to rise, while the opposite trend is seen in men [79]. However, in patients with manic depressive disorder, no significant gender-related differences in plasma nesfatin-1 levels have been detected [62,68].

The current study’s findings, as is consistent with previous research results [29,80], reveal alterations in plasma serotonin, dopamine, and oxytocin levels, all recognized for their involvement in mediating aggressive behavior. Under normal conditions, serotonin exerts its influence on the frontal regions of the brain, where it dampens amygdala activity, a critical component of the limbic system, which is responsible for regulating emotional responses like fear and anger. Consequently, serotonin promotes a sedative effect, and reduced serotonin levels correlate with uncontrolled impulsive and aggressive behaviors [81]. Serotonin’s regulation extends to the prefrontal cortex, where lower serotonin levels influence responses to external stimuli, heightening susceptibility to aggression and diminishing emotional control. Anticipating risks becomes challenging, prompting impulsive engagement in aggressive behavior. In support of this, research has linked deficient serotonergic function to mouse-killing behavior in rodents [82]. Similarly, non-human primates with reduced serotonin levels exhibit increased impulsive and aggressive behaviors [83,84]. Low concentrations of serotonin in cerebral spinal fluid have also been associated with poor impulse control and aggression in adolescent monkeys [85]. Moreover, female primates displaying impulsivity and risky behavior have been found to possess low serotonin concentrations in the cerebral spinal fluid [86].

The dopaminergic system is responsible for behavioral activation, motivated behavior, and reward processing, and actively modulates aggressive behaviors [87,88]. Increased dopaminergic system activity has been consistently linked to impulsive aggression in animal studies [89,90]. Investigations into aggressive behavior in rodents have consistently reported elevated dopamine levels before, during, and after aggressive encounters [91,92,93]. Additionally, serotonergic function deficiency may result in hyperactivity of the dopamine system, further promoting aggressive behavior [94].

Oxytocin plays a pivotal role in stress and aggression, as evidenced by animal studies that have linked it to maternal behavior, aggression, non-social behaviors [95], and the regulation of stress responses [96]. Interestingly, a history of aggression has been found to inversely correlate with oxytocin levels in cerebral spinal fluid, suggesting that oxytocin plays a mechanistic role in modulating aggressive behavior [97]. The total oxytocin receptor-knockout male mice had heightened aggression compared with the controls, while the predominantly forebrain-specific oxytocin receptor-knockout male mice displayed similar aggression levels to control mice [98]. This animal study indicates that oxytocin may be essential to developing neural circuits that underlie aggression in adulthood.

The decrease in plasma serotonin and oxytocin levels coupled with an increase in plasma dopamine levels, which were observed in our study as a consequence of aggressive behavior, align with previous research [34,90,97] on the roles of these neuropeptides in aggression. These findings suggest a potential involvement of these neuropeptides in aggressive tendencies in dogs.

An intricate interplay exists at the central nervous system level among key neurotransmitters, namely, serotonin, dopamine, oxytocin, and nesfatin-1, all recognized for their significance in regulating aggressive behavior. Nesfatin-1 has been reported to be co-expressed with serotonin and oxytocin in the central nervous system [49,99]. In addition, nesfatin-1 shows its physiological effects by using the receptors of serotonergic and oxytocinergic systems and interacting with them through multiple projections. Notably, peripheral administration of a serotonin 5-HT receptor antagonist in rodents has been shown to stimulate hypothalamic nesfatin-1 secretion [100]. Furthermore, nesfatin-1 has a direct depolarizing effect on oxytocinergic neurons and, when centrally administered, activates both magnocellular and parvocellular oxytocin neurons, ultimately stimulating oxytocin release [101,102]. Additionally, dopamine neurons in the brain’s ventral tegmental area express nesfatin-1 [47], and nesfatin-1 has been found to modulate dopamine neuron activity in regions such as the substantia nigra and ventral tegmental area [103]. The results of our study confirm the interactions of these peptides in both central and peripheral mechanisms and their effects on changes in aggression and feeding states.

## 5. Conclusions

In conclusion, the findings from the present study reveal that serotonin, dopamine, oxytocin, and nesfatin-1 play significant roles in aggression in Pit Bull dogs, which are known for their predisposition to aggression. These findings underscore the intricate interactions of these neuropeptides within the central nervous system, illuminating the complex mechanisms governing aggressive behavior in canine. Notably, our findings introduce a novel perspective on the role of nesfatin-1 in aggressive behavior, particularly in dogs. In addition, our data strengthen the sparse data on plasma nesfatin-1 in aggression-related psychotic diseases in humans, showing that the effect of inducing aggression in dogs is devoid of cognitive complexity. While our study exclusively concentrated on Pit Bull dogs, the findings establish a fundamental understanding of the neurobiological foundations of canine aggression, present potential insights into innovative approaches for managing aggression, and lay the groundwork for developing novel strategies for aggression treatment.

## Figures and Tables

**Figure 1 animals-14-00632-f001:**
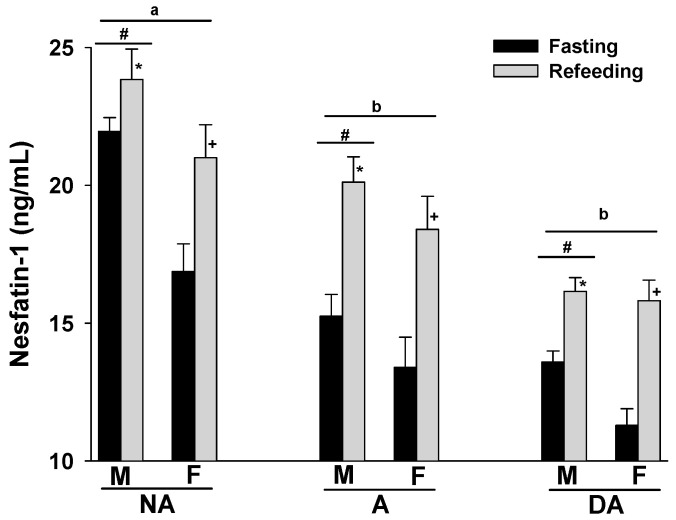
Plasma nesfatin-1 levels of Pit Bulls categorized by hunger/satiety status, gender, and severity of aggression. The data are presented as mean ± SEM of measurements. Statistical analysis was performed using ANOVA and Tukey’s HSD to evaluate differences in plasma nesfatin-1 levels among groups (NA, A, and DA). Paired *t*-tests were utilized to compare the variances in nesfatin-1 levels between fasted and fed states, as well as between female and male dogs. A value of *p* < 0.05 was considered statistically significant. The annotations ‘a’ and ‘b’ indicate the differences in nesfatin-1 levels among aggression groups; ‘#’ represents differences in nesfatin-1 levels between genders; ‘*’ illustrates differences in nesfatin-1 levels between male dogs in hunger–satiety states, and ‘+’ differences in nesfatin-1 levels between female dogs in hunger–satiety states. Key abbreviations: F, fasting; R, refeeding; M, male; F, female; NA, non-aggressive; A, aggressive; DA, dominant aggressive.

**Figure 2 animals-14-00632-f002:**
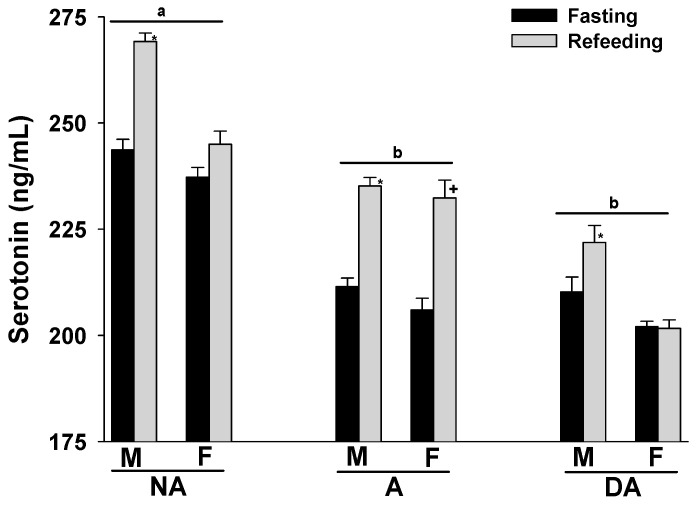
Plasma serotonin levels of Pit Bulls categorized by hunger/satiety status, gender, and severity of aggression. The data are presented as mean ± SEM of measurements. Statistical analysis was performed using ANOVA and Tukey’s HSD to assess differences in plasma serotonin levels among groups (NA, A, and DA). Paired *t*-tests were utilized to compare the variances in serotonin levels between fasted and fed states, as well as between female and male dogs. A value of *p* < 0.05 was considered statistically significant. The annotations ‘a’ and ‘b’ indicate the differences in serotonin levels among aggression groups; ‘*’ illustrates the differences in serotonin levels between male dogs in hunger–satiety states, and ‘+’ differences in serotonin levels between female dogs in hunger–satiety states. Key abbreviations: F, fasting; R, refeeding; M, male; F, female; NA, non-aggressive; A, aggressive; DA, dominant aggressive.

**Figure 3 animals-14-00632-f003:**
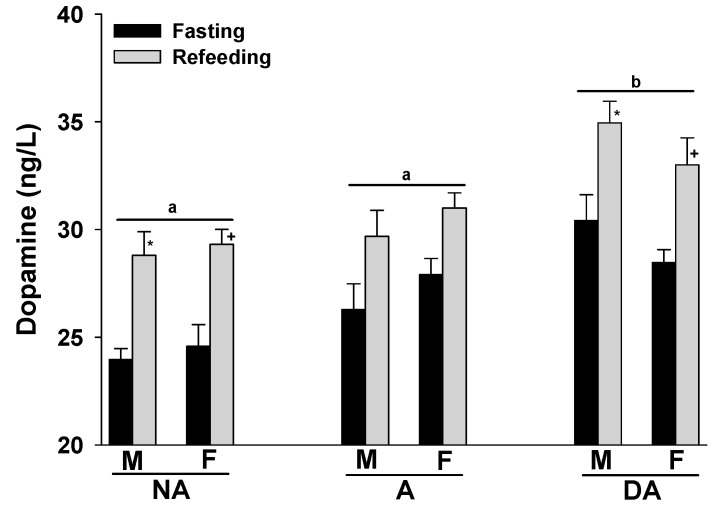
Plasma dopamine levels of Pit Bulls categorized by hunger/satiety status, gender and severity of aggression. The data are presented as mean ± SEM of measurements. Statistical analysis was performed using ANOVA and Tukey’s HSD to evaluate differences in plasma dopamine levels among groups (NA, A, and DA). Paired *t*-tests were utilized to compare the variances in dopamine levels between fasted and fed states, as well as between female and male dogs. A value of *p* < 0.05 was considered statistically significant. The annotations ‘a’ and ‘b’ indicate the differences in dopamine levels among aggression groups; ‘*’ illustrates the differences in dopamine levels between male dogs in hunger–satiety states, and ‘+’ differences in dopamine levels between female dogs in hunger–satiety states. Key abbreviations: F, fasting; R, refeeding; M, male; F, female; NA, non-aggressive; A, aggressive; DA, dominant aggressive.

**Figure 4 animals-14-00632-f004:**
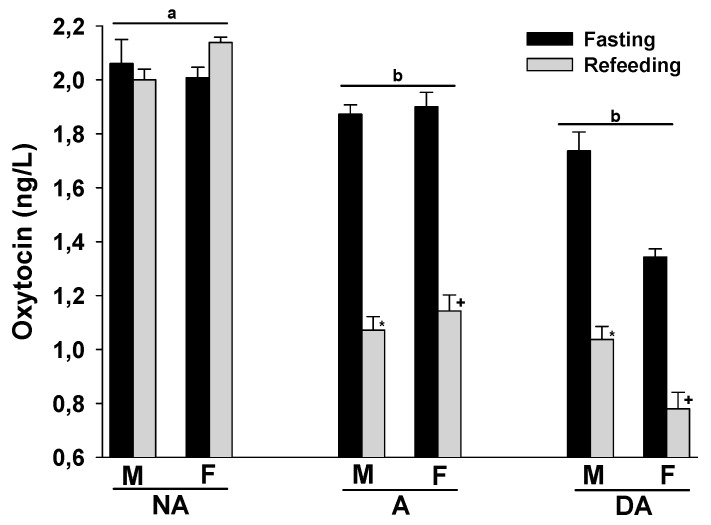
Plasma oxytocin levels of Pit Bulls categorized by hunger/satiety status, gender, and severity of aggression. The data are presented as mean ± SEM of measurements. Statistical analysis was performed using ANOVA and Tukey’s HSD to evaluate differences in plasma oxytocin levels among groups (NA, A, and DA). Paired *t*-tests were utilized to compare the variances in dopamine levels between fasted and fed states, as well as between female and male dogs. A value of *p* < 0.05 was considered statistically significant. The annotations ‘a’ and ‘b’ indicate the differences in oxytocin levels among aggression groups; ‘*’ illustrates the differences in oxytocin levels between male dogs in hunger–satiety states, and ‘+’ differences in oxytocin levels between female dogs in hunger–satiety states. Key abbreviations: F, fasting; R, refeeding; M, male; F, female; NA, non-aggressive; A, aggressive; DA, dominant aggressive.

## Data Availability

The data presented in this study are available upon request from the corresponding author.

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
