# Peer review of "The Relation between Plasma Nesfatin-1 Levels and Aggressive Behavior in Pit Bull Dogs"

_animals, 2024, doi:10.3390/ani14040632_

Round 1
Reviewer 1 Report
Comments and Suggestions for Authors
Dear Authors,
It's an interesting article. Its main contribution is based on the direct relationship between low levels of Nesfatin-1 and aggressiveness. It helps, therefore, to analyze the different functions of nesfatin-1.
If I may, I would like to make a few observations regarding your publication:
1.- Modern ethologists can question this classical view of aggressiveness due to dominance since they understand that the problems of interspecific aggressiveness (towards the family, for example) would not be a classic hierarchical conflict for dominance but would be related "to the protection of valuable resources or the defense against manipulations perceived as threatening to the dog. where the animal's difficulty in predicting what is going to happen, the frustration of expectations, learning and fear, especially if there are exaggerated punishments, play a determining role.
2.-Regarding the Quantification Scale used, is it validated?
On the other hand, in his bibliographical citation (MacNeil-Allcock, A.; Clarke, N.M.; Ledger, R.A.; Fraser, D. Aggression, behaviour, and animal care among pit bulls and other 436 dogs adopted from an animal shelter. Anim. Welf. 2011, 20, 463-468) mentions the use of the Aggression Quantification Scale in study dogs. For this study to be reproducible, don't you think the work of Sherman et al. (1996) should be included? (It would be the following reference: Sherman C, Reisner I, Taliaferro L and Houpt K 1996 Characteristics, treatment, and outcome of 99 cases of aggression between dogs. Applied Animal Behaviour Science 47: 91-108.)
3.-The processing of the results by SPSS, the figures used, and the discussion are clearly shown and very revealing.
4.-It may be appropriate to include future lines of study such as: "In future research it will be necessary to evaluate the variations of nesfatin-1 over time and evaluate the response to different pharmacological treatments in dogs.
MINOR ASPECTS:
1.-[133] 1.-[133] Correct citation 34 (Landsberg, G.M. Diagnosing dominance agresyon. Can. Vet. J. 1990, 31, 45-46. 433. Change “agresyon” by aggression.
2.-Review the rest of the bibliography to ensure that it is correct at the time of publication.
3.-In [127], where the name of the laboratory (Bioassay Technology Laboratory) is included, consider the inclusion of the city and the country (Shanghai, China?).
4.-If you are so kind, check the identification in the catalog numbers of the ELISA kits used. I have tried to review it in https://fybreeds.com/userfiles/files/BioAssay%20Product%20List1.pdf, but it only includes in this catalog the number of the nesfatin-1. The catalog numbers for serotonin, dopamine and oxytocin would need to be checked again. Thank you!
Author Response
Dear Reviewer,
We express our sincere gratitude for the attention you dedicated to our study and the time you invested in providing valuable feedback. Your positive and constructive comments have been instrumental in enhancing the manuscript. We have incorporated revisions based on your remarks, aiming to make our work more accessible and appealing to the readership.
We appreciate your positive feedback and constructive criticism, which we believe has significantly improved the overall quality of our manuscript. Below, you will find responses to your questions in accordance with the comments and suggestions you provided.
Comments 1: Modern ethologists can question this classical view of aggressiveness due to dominance since they understand that the problems of interspecific aggressiveness (towards the family, for example) would not be a classic hierarchical conflict for dominance but would be related "to the protection of valuable resources or the defense against manipulations perceived as threatening to the dog. where the animal's difficulty in predicting what is going to happen, the frustration of expectations, learning and fear, especially if there are exaggerated punishments, play a determining role.
Response 1: In our country, the care of Pit Bull breed dogs is quite prevalent in certain circles. We are aware that individuals striving to portray a powerful image often capitalize on the characteristic phenotypes of these dogs, taking on the responsibility for the care of such breeds. While some owners provide a peaceful home environment for these animals, unfortunately, it is also common for these dogs to be raised with a focus on aggression. Individuals attempting to assert themselves as dominant often subject these dogs to undesirable conditions while ensuring their care. Regrettably, these practices include feeding them raw meat in dark rooms and exposing them to loud noises, triggering fears without knowing the potential consequences for the animals. Some individuals resort to distressing practices such as applying exaggerated punishments and inducing aggression through explicit violence. Animals exposed to these practices are, unfortunately, often walked on the streets without muzzles or leashes, posing a threat to both people and other animals in the vicinity. These unsettling practices underscore the need for increased awareness and responsible pet ownership.
In Turkey, the Animal Protection Law No. 5199, enacted in 2004, prohibits the production, adoption, rehoming, housing, feeding, exchange, display, gifting, and sale of Pit bulls. In 2021, the Law on the Amendment of the Animal Protection Law and the Turkish Penal Code was published in the Official Gazette. As per Article 14/1 of this law, individuals engaged in the aforementioned activities are subject to administrative fines. The same article also forbids the transportation of "dangerous" animals registered before January 14, 2022, without a registration certificate, muzzle, and collar, as well as their entry into public areas and children's playgrounds/parks. Following the enactment of the law, owners of dogs classified under the aggressive breed category have either released these animals onto the streets or abandoned them in shelters to evade associated costs. These dogs, still under state protection, continue to be cared for in shelters, and adoption is currently prohibited.
In the aggression test we employed in our study, we included questions to gather information about the history of each dog. The shelter staff and veterinary professionals, who have access to all information since the dogs' arrival at the shelter, assisted us in obtaining comprehensive details. The aggression test, individually administered to each dog, allowed us to acquire detailed information about each canine participant. We were able to obtain information such as whether the dog had been abandoned on the street or left at the shelter by its owner, whether it had experienced previous violence, whether it had harmed other animals or humans, whether it had been involved in dog fighting, or if it had exhibited known aggressive behaviors. If the dog was not left at the shelter by its owner and we couldn't obtain detailed information about the dog's history, we relied on information gathered from people in the area where the dog was found and details provided by dog rescue teams for street-abandoned dogs. The aggression test we conducted aimed to evaluate the degrees of aggression displayed by dogs in situations such as guarding their food, protecting their territory, self-defense due to the inability to predict the outcome based on past experiences when touched, defending against unwanted situations like loud noises, protecting familiar individuals from strangers, or self-defense against potentially punitive actions like raising a hand, rather than determining classical hierarchical aggression for dominance. The test focused on assessing the aggression levels exhibited by the animals in response to these scenarios, including behaviors such as growling/barking, displaying teeth, distant biting gestures, jumping, attempting to bite, etc.
Comments 2: Regarding the Quantification Scale used, is it validated?
On the other hand, in his bibliographical citation (MacNeil-Allcock, A.; Clarke, N.M.; Ledger, R.A.; Fraser, D. Aggression, behaviour, and animal care among pit bulls and other 436 dogs adopted from an animal shelter. Anim. Welf. 2011, 20, 463-468) mentions the use of the Aggression Quantification Scale in study dogs. For this study to be reproducible, don't you think the work of Sherman et al. (1996) should be included? (It would be the following reference: Sherman C, Reisner I, Taliaferro L and Houpt K 1996 Characteristics, treatment, and outcome of 99 cases of aggression between dogs. Applied Animal Behaviour Science 47: 91-108.)
Response 2: The aggression test we applied to the dogs is based on the C-BARQ test, developed and validated by Yuying Hsu and James Serpell in 2003. This test has been publicly available through an online platform since 2005. The references used for the aggression test in our study are rooted in the principles of the C-BARQ test. The C-BARQ test is designed to measure, classify, and analyze various aspects of dog behavior, including general behavioral tendencies, social compatibility, fear responses, aggressive behaviors, and other traits. The survey covers a wide range of behavioral features, such as social interactions, aggression, fears and phobias, energy levels, and play tendencies. Given that our study is based on the C-BARQ test, there are no concerns about validation. In our shelter-based study, as the dogs were individually housed in cages without direct contact with other animals and humans, we modified and applied the C-BARQ test accordingly. Additionally, we omitted questions related to sub-dimensions that were not suitable for our shelter sample, and we did not assess handler compatibility, considering the difficulty of implementing such measures in a shelter environment, where animals are generally under stress. We aimed to avoid practices that could potentially increase stress factors for the animals. The section 2.1 has been revised in the manuscript to eliminate any confusion in meaning.
Comments 3: The processing of the results by SPSS, the figures used, and the discussion are clearly shown and very revealing.
Response 3: Thank you for your valuable feedback.
Comments 4: It may be appropriate to include future lines of study such as: "In future research it will be necessary to evaluate the variations of nesfatin-1 over time and evaluate the response to different pharmacological treatments in dogs.
Response 4: We have added to the Conclusions section of our study, acknowledging your insights, that our research may lay potential groundwork for future studies and new treatment strategies. Thank you for your contribution.
MINOR ASPECTS:
Comments 5: [133] 1.-[133] Correct citation 34 (Landsberg, G.M. Diagnosing dominance agresyon. Can. Vet. J. 1990, 31, 45-46. 433. Change “agresyon” by aggression.
Response 5: The spelling mistake in the manuscript has been corrected.
Comments 6: Review the rest of the bibliography to ensure that it is correct at the time of publication.
Response 6: All bibliographies have been reviewed once again.
Comments 7: In [127], where the name of the laboratory (Bioassay Technology Laboratory) is included, consider the inclusion of the city and the country (Shanghai, China?).
Response 7: The city and country information for the company from which the ELISAs were obtained has been added to the manuscript.
Comments 8: If you are so kind, check the identification in the catalog numbers of the ELISA kits used. I have tried to review it in https://fybreeds.com/userfiles/files/BioAssay%20Product%20List1.pdf, but it only includes in this catalog the number of the nesfatin-1. The catalog numbers for serotonin, dopamine and oxytocin would need to be checked again. Thank you!
Response 8: When I rechecked the current kit list from BT Lab, unfortunately, I couldn't find any kits other than Canine Nesfatin-1, indicating that they may have discontinued production. Our study period covers February to June 2023, and we have the user instructions for the kits. The kit manuals, including catalog numbers, can be shared at any time upon request. Additionally, I found an old catalog, which you can review below if you're interested.
Thank you once again for your thorough review and invaluable input.
Sincerely,
Reviewer 2 Report
Comments and Suggestions for Authors
This is an exciting line of research that will be of great interest to readers. The authors have done a nice job describing the relevant neurobiological foundations for aggressive behavior and the methodology used for analyzing the plasma samples appears sound.
That said, I have serious concerns about other aspects of the methods that I believe prevent publication at this time. I'm hopeful that the authors can address these issues and publish their work. I respectfully suggest that, if they haven't already, that they connect with someone who studies animal behavior to assist with revisions.
* No definition of "pit bull" dogs is provided. It is widely accepted that the term "pit bull" refers to a variety of recognized breeds, and so more information is needed about the subjects of the study.
* It is also widely accepted that breed determination cannot be made simply by visual inspection. I understand that conducting genetic testing on the dogs would likely not be possible or conclusive, but the authors must go into much more detail about the dogs that were chosen for the study and why there were chosen.
* No operational definition of aggression is provided.
* Far more detail is required about the behavioral assessment that occurred that led to the classification of the dogs. Without it, there is a huge gap in the manuscript.
* Also related to the behavioral assessment is that there is no evidence that tests of inter-observer reliability occurred. I understand why this could be the case, but it needs to be addressed.
* The focus on the study is "pit bull" dogs, and while this is understandable, it leaves me wondering what the same measures would be in different breeds of dogs. Without a group to which the results can be compared, they have far less value. Perhaps a more in-depth discussion of references 3 and 4 would assist here.
I also ask that the authors use caution with the language that they use. Most notably, in line 172: "Aggression resulted in an increase in plasma dopamine levels". It isn't accurate to say that aggression resulted in that increase. It seems more accurate to state that the authors observed a correlation/relationship between the two variables.
Comments on the Quality of English LanguageEnglish language is very good, and this is a well-written manuscript.
Author Response
Dear Reviewer,
We sincerely appreciate your positive and constructive feedback as well as your valuable contributions. Additionally, we would like to express our gratitude for the valuable time you dedicated to our study. We believe that your comments will enhance the appeal of our study to readers and increase its overall interest. We have made the suggested corrections to the best of our abilities in the manuscript. Below, you can find responses to your questions based on your comments and critiques.
First and foremost, as you suggested, we established connections with researchers focusing on animal behavior, particularly those working with dogs. With their assistance, we have made the necessary adjustments in our manuscript.
Comments 1: No definition of "pit bull" dogs is provided. It is widely accepted that the term "pit bull" refers to a variety of recognized breeds, and so more information is needed about the subjects of the study.
Response 1: Explanations regarding the term "Pit Bull" have been added to the introduction section of the manuscript.
Comments 2: It is also widely accepted that breed determination cannot be made simply by visual inspection. I understand that conducting genetic testing on the dogs would likely not be possible or conclusive, but the authors must go into much more detail about the dogs that were chosen for the study and why there were chosen.
Response 2: We greatly appreciate your valuable feedback. We are aware that genetic tests provide precise and accurate results in determining the breed, and relying solely on visual examination may not be sufficient. The funding for our study is provided by our university, and unfortunately, the budget we received covers only the expenses for consumables. Due to the economic challenges within our country and the fluctuations in the exchange rates of the Dollar/Euro, we had to work with a very modest budget. If it were feasible, we would have preferred to categorize the animals genetically as well. We selected the dogs included in our study based on the phenotypic characteristics of pit bull dogs determined in Levine and Poray-Wybranowska's study (2016). Additionally, until January 2022, the production and adoption of pit bull breed dogs in our country were not problematic. Pitbull dogs were highly valued in our country and owning them was seen as a positive symbol by some groups. Therefore, until January 2022, the adopted pit bull breed dogs were carefully selected, and their pedigrees were well-documented, emphasizing the importance of them being purebred. In Turkey, the Animal Protection Law No. 5199, enacted in 2004, prohibits the production, adoption, rehoming, housing, feeding, exchange, display, gifting, and sale of Pit bulls. In 2021, the Law on the Amendment of the Animal Protection Law and the Turkish Penal Code was published in the Official Gazette. As per Article 14/1 of this law, individuals engaged in the aforementioned activities are subject to administrative fines. The same article also forbids the transportation of "dangerous" animals registered before January 14, 2022, without a registration certificate, muzzle, and collar, as well as their entry into public areas and children's playgrounds/parks. Following the enactment of the law, owners of dogs classified under the aggressive breed category have either released these animals onto the streets or abandoned them in shelters to evade associated costs. These dogs, still under state protection, continue to be cared for in shelters, and adoption is currently prohibited. The dogs included in our study consist of those either abandoned by their owners to shelters or seized by the government due to their owners not fulfilling the necessary procedures. In the aggression test we employed in our study, we included questions to gather information about the history of each dog. The shelter staff and veterinary professionals, who have access to all information since the dogs' arrival at the shelter, assisted us in obtaining comprehensive details. According to the information obtained, there are no mixed breeds among the dogs we used in our study. Enlightening information regarding the subject has been added to the manuscript.
Ref: Levine, R., Poray-Wybranowska, J. 2016. American Bully: Fear, Paradox, and the New Family Dog. Otherness: Essays and Studies 5.2.
Comments 3: No operational definition of aggression is provided.
Response 3: The categorization of aggressive behaviors exhibited by dogs in response to behaviors such as food guarding, territory guarding, self-protection against unforeseen influences, and protection of familiar individuals (Quiet and calm; Growling and barking; Showing teeth; Biting from a distance; Aggressive attack/Biting) has been added to the manuscript.
Comments 4: Far more detail is required about the behavioral assessment that occurred that led to the classification of the dogs. Without it, there is a huge gap in the manuscript.
Response 4: In line with your feedback, additional information about both the aggression test and the evaluation of the test has been added to the Materials and Methods section of the manuscript.
Comments 5: Also related to the behavioral assessment is that there is no evidence that tests of inter-observer reliability occurred. I understand why this could be the case, but it needs to be addressed.
Response 5: The aggression test administered to the dogs is based on the C-BARQ test developed and validated by Yuying Hsu and James Serpell in 2003. This test has been publicly available through an online platform since 2005. The C-BARQ test is designed to measure, classify, and analyze various behavioral aspects of dog behavior, encompassing general behavioral tendencies, social interactions, fear responses, aggressive behaviors, and other characteristics. It covers a wide range of behavioral features, including social interactions, aggression, fears and phobias, energy levels, play tendencies, and more. Although the C-BARQ test is typically applicable to all dogs and widely used by dog owners and behavior scientists, we modified its application in the shelter setting. Due to the dogs being housed individually in cages without direct contact with other animals or people, we made adjustments to the C-BARQ test. Additionally, we refrained from using questions that were not suitable for our sample in the C-BARQ test and omitted assessments for observer compatibility. Given the challenging shelter environment and the general stress experienced by the animals, we opted to avoid practices that could potentially increase stress factors.
Comments 6: The focus on the study is "pit bull" dogs, and while this is understandable, it leaves me wondering what the same measures would be in different breeds of dogs. Without a group to which the results can be compared, they have far less value. Perhaps a more in-depth discussion of references 3 and 4 would assist here.
Response 6: While our study primarily focuses on Pit Bull breed dogs, we believe that our findings highlight the significance of neuropeptides in dog aggression, which could potentially play a role in aggression across various dog breeds. In response to your feedback, we have included a comparison of aggression levels in other dog breeds in the manuscript. We intentionally chose to concentrate on Pit Bull breed dogs because of the prevalent negative news and discussions about their aggressiveness, posing a perceived threat to people and other dogs, especially in our country. Investigating aggression in these dogs, recently categorized as aggressive breeds, and shedding light on the role of neuropeptides in their aggression, even hormonally, is considered a valuable contribution to the literature. We acknowledge that attributing aggression solely to one breed is not feasible. However, delving into the mechanisms in breeds known for their aggression, like Pit Bulls, may provide insights into similar mechanisms in other breeds. Our study, being the first to suggest a potential role of nesfatin-1 in aggression in humans and other animals, is believed to make significant contributions to the literature. While our method alone cannot fully reveal this relationship, our study lays the groundwork for more in-depth molecular and genetic research to illuminate the mechanisms of aggression
Comments 7: I also ask that the authors use caution with the language that they use. Most notably, in line 172: "Aggression resulted in an increase in plasma dopamine levels". It isn't accurate to say that aggression resulted in that increase. It seems more accurate to state that the authors observed a correlation/relationship between the two variables.
Response 7: We appreciate your feedback. In line with your suggestions, necessary revisions have been made in the results and discussion sections of the manuscript.
Thank you once again for your insightful feedback, which has undoubtedly improved the quality of our work.
Best regards,
Reviewer 3 Report
Comments and Suggestions for Authors
Aggression can be considered a behavioral problem when it is excessive or inappropriate to the situation. In addition, the reasons why dogs present aggression are well identified. Attributing aggressive behavior to a specific breed is quite risky, because often this behavior is caused by an abnormal human-dog relationship. Nesfatin- 1 is an appetite suppressant and gastroprotective peptide and not associated with aggression. The study methodology does not guarantee results indicative of excessive aggression.
Author Response
Dear Reviewer,
Thank you for your valuable feedback and the time you have invested in reviewing our manuscript. We appreciate your insights and would like to address the concerns you raised regarding our study.
Reviewer: Aggression can be considered a behavioral problem when it is excessive or inappropriate to the situation. In addition, the reasons why dogs present aggression are well identified. Attributing aggressive behavior to a specific breed is quite risky, because often this behavior is caused by an abnormal human-dog relationship. Nesfatin-1 is an appetite suppressant and gastroprotective peptide and not associated with aggression. The study methodology does not guarantee results indicative of excessive aggression.
Authors: Firstly, we concur with your observation that aggression in dogs can stem from a multitude of factors, including aspects of the human-dog relationship, environmental influences, and training methods. We acknowledge the importance of avoiding breed-specific generalizations and fully recognize the complexity inherent in canine behavior. Your emphasis on this aspect reinforces the need for a nuanced understanding of aggression in dogs, which we strive to reflect in our research.
Regarding your comments on nesfatin-1, we appreciate the clarification that it is predominantly known as an appetite suppressant and gastroprotective peptide. Our study sought to explore the potential connections between various neuropeptides and canine aggression, given the multifaceted nature of such behaviors. While nesfatin-1’s primary functions are well-documented, our intention was to investigate whether there might be secondary or less-explored roles of this peptide in the context of canine aggression, contributing to the broader research in this field.
As for the methodology of our study, we understand and acknowledge the complexities involved in accurately studying and interpreting aggression in dogs. We agree that our approach, while designed to be as thorough and rigorous as possible, has limitations and cannot offer definitive results. Our goal was not to provide conclusive evidence but rather to add to the ongoing discourse on canine aggression, particularly regarding potential neurobiological correlates. We believe that our research can serve as a steppingstone for future studies in this area, and we are committed to refining our methodologies in light of new insights and feedback such as yours.
We hope that our responses address your concerns adequately, and we are open to further suggestions or guidance that could enhance the quality and impact of our research.
Thank you once again for your review and comments.
Best regards,
Round 2
Reviewer 2 Report
Comments and Suggestions for Authors
Thank you for the time you spent revising this manuscript. It is much improved and will have value to the animal sheltering community. I noticed in at least one place in the manuscript that the same sentence was included twice. I suspect this is just a result of the revision process and that this situation, and any others like it, will be addressed in the editing process.
Comments on the Quality of English LanguageQuality of English language is good.
Author Response
Dear Reviewer,
Thank you for your positive feedback. We have incorporated the suggested revisions into the manuscript and highlighted them in yellow.
Thank you once again for the valuable time you spent evaluating our manuscript.
Kind regards.
Reviewer 3 Report
Comments and Suggestions for Authors
The explanations provided by the authors and the proofreading of parts of the manuscript are, in my opinion, sufficient. I recommend the article for publication.
Author Response
Dear Reviewer,
Thank you for your positive feedback. Thank you once again for the valuable time you spent evaluating our manuscript.
Kind regards.